# Geographical Differences and the National Meeting Effect in Patients with Out-of-Hospital Cardiac Arrests: A JCS–ReSS Study Report

**DOI:** 10.3390/ijerph16245130

**Published:** 2019-12-16

**Authors:** Tetsuya Yumoto, Hiromichi Naito, Takashi Yorifuji, Yoshio Tahara, Naohiro Yonemoto, Hiroshi Nonogi, Ken Nagao, Takanori Ikeda, Naoki Sato, Hiroyuki Tsutsui

**Affiliations:** 1Department of Emergency, Critical Care, and Disaster Medicine, Okayama University Graduate School of Medicine, Dentistry and Pharmaceutical Sciences, 2-5-1 Kita-ku, Shikata-cho, Okayama 700-8558, Japan; 2Department of Epidemiology, Okayama University Graduate School of Medicine, Dentistry and Pharmaceutical Sciences, 2-5-1 Kita-ku, Shikata-cho, Okayama 700-8558, Japan; 3Department of Cardiovascular Medicine, National Cerebral and Cardiovascular Center, 6-1 Kishibe-Shimmachi, Suita, Osaka 564-8565, Japan; 4National Center of Neurology and Psychiatry, 4-1-1 Ogawa-Higashi, Kodaira, Tokyo 187-8551, Japan; 5Intensive Care Center, Shizuoka General Hospital, 4-27-1 Kitaando, Aoiku, Shizuoka 420-8527, Japan; 6Cardiovascular Center, Nihon University Hospital, 1-8-13 Kanda Surugadai, Chiyoda-ku, Tokyo 101-8309, Japan; 7Department of Cardiovascular Medicine, Toho University Faculty of Medicine, 5-21-16 Omorinishi, Ota-ku, Tokyo 143-8540, Japan; 8Cardiovascular Medicine, Kawaguchi Cardiovascular and Respiratory Hospital, 1-1-51 Maekawa Kawaguchi-shi, Saitama 333-0842, Japan; 9Department of Cardiovascular Medicine, Kyushu University Faculty of Medical Sciences, 3-1-1 Maidashi, Higashi-ku, Fukuoka 812-8582, Japan

**Keywords:** out-of-hospital cardiac arrest, outcome, national meeting, cardiopulmonary resuscitation

## Abstract

The “national meeting effect” refers to worse patient outcomes when medical professionals attend academic meetings and hospitals have reduced staffing. The aim of this study was to examine differences in outcomes of patients with out-of-hospital cardiac arrest (OHCA) admitted during, before, and after meeting days according to meeting location and considering regional variation of outcomes, which has not been investigated in previous studies. Using data from a nationwide, prospective, population-based, observational study in Japan, we analyzed adult OHCA patients who underwent resuscitation attempts between 2011 and 2015. Favorable one-month neurological outcomes were compared among patients admitted during the relevant annual meeting dates of three national scientific societies, those admitted on identical days the week before, and those one week after the meeting dates. We developed a multivariate logistic regression model after adjusting for confounding factors, including meeting location and regional variation (better vs. worse outcome areas), using the “during meeting days” group as the reference. A total of 40,849 patients were included in the study, with 14,490, 13,518, and 12,841 patients hospitalized during, before, and after meeting days, respectively. The rates of favorable neurological outcomes during, before, and after meeting days was 1.7, 1.6, and 1.8%, respectively. After adjusting for covariates, favorable neurological outcomes did not differ among the three groups (adjusted OR (95% CI) of the before and after meeting dates groups was 1.03 (0.83–1.28) and 1.01 (0.81–1.26), respectively. The “national meeting effect” did not exist in OHCA patients in Japan, even after comparing data during, before, and after meeting dates and considering meeting location and regional variation.

## 1. Introduction

Out-of-hospital cardiac arrest (OHCA) is a significant public health problem worldwide [1]. In Japan, over 127,000 OHCAs occur annually and one-month survival rates remain low despite improvements in healthcare, including public education, emergency medical systems, and integrated post-cardiac arrest care [2,3,4]. Thorough efforts have been undertaken to improve patient outcomes [5].

Adequate medical staffing and resource allocation is important to ensure that high-quality care is provided to patients. Off-hours presentation has been associated with worse outcomes among patients with OHCA, acute myocardial infarction (AMI), and those hospitalized in the intensive care unit [6,7,8,9]. It is speculated that this “weekend effect” occurs due to insufficient medical staffing [6,7,8,9]. Similarly, a “national meeting effect”, or differences in clinical outcomes among OHCA or AMI patients admitted during relevant national academic meeting dates and nonmeeting dates, has been investigated [10,11,12]. In the United States, a lower adjusted 30-day mortality was observed among patients after being OHCA-hospitalized in major teaching hospitals during national cardiology meeting dates, while no significant differences in one-month survival and favorable neurological outcomes were found in OHCA patients admitted during national academic meeting dates and nonmeeting dates in Japan [10,11].

One of the limitations of these studies was that the control group was defined as patients admitted with OHCA during the same calendar days in the weeks before and after the meetings [10,11,12]. Furthermore, regional differences should have been considered because regional variation in patient outcomes exists and would differ by national meeting location [13,14]. We hypothesized that patient outcomes would be different taking these variables into consideration. This study aimed to examine the hypothesis that outcomes after OHCA would differ among patients admitted before, during, and after meeting dates according to the geographical region where they were treated and the location where the national academic meeting had been held using a different study period from the previous one.

## 2. Materials and Methods

### 2.1. Study Design and Data Sources

This study was conducted using the All-Japan Utstein Registry of the Fire and Disaster Management Agency (FDMA), consisting of nationwide population-based data on OHCA patients compiled in accordance with Utstein-style data collection [15], the details of which have been described elsewhere [16]. We defined cardiac arrest as the stoppage of mechanical cardiac activities determined by the absence of circulation signs [16]. Cause of arrest was presumed to be cardiac unless evidence of a noncardiac origin, such as external causes (trauma, asphyxia, drug overdose, drowning, or hanging), cerebrovascular disease, malignant tumor, respiratory disease, or any other noncardiac cause, could be clearly suggested. Attending physicians determined cardiac vs. noncardiac diagnoses in cooperation with emergency medical services (EMS) personnel. The Okayama University Hospital ethical committee approved this study (ID: 1904-002). The requirement for informed consent was waived because of the anonymous nature of the data.

### 2.2. EMS System in Japan

In 2015, Japan had a population of approximately 127 million people in a 378,000 km^2^ area [17]. Details of the EMS system in Japan have been previously reported elsewhere [3]. Briefly, in 2015, there were 750 fire stations with dispatch centers nationwide, which currently provide regional 24/7 EMS [18]. Generally, three EMS staff members, including at least one emergency life-saving technician (ELST), are dispatched to the scene when people call an ambulance. ELSTs can place peripheral intravenous lines, insert supraglottic airway devices, or use semi-automated external defibrillators. Specially trained ELSTs are authorized to perform endotracheal intubation and give patients intravenous epinephrine. All EMS providers conduct cardiopulmonary resuscitation (CPR) for OHCA patients according to Japanese CPR guidelines, which are based on the International Liaison Committee on Resuscitation [19].

### 2.3. Choice of Hospital and Post-Arrest Care after OHCA

EMS personnel at the scene contact an appropriate hospital nearby according to local protocols to request admission of the OHCA patient based on their condition. Receiving hospitals provide resuscitation and emergency care after hospital arrival and post-arrest care, including percutaneous coronary intervention and therapeutic hypothermia according to the attending physician in charge and local practices.

### 2.4. Data Collection and Quality Control

Based on the Utstein-style template [15], the following data were obtained: Age, sex, date of hospital arrival, prefecture where the OHCA events occurred, bystander witness status, type of bystander witness, initial documented cardiac rhythm, etiology of cardiac arrest, time course of resuscitation, bystander-initiated CPR, use of public-access automated external defibrillation (AED), advanced airway management, and prehospital administration of intravenous fluids and epinephrine, as well as outcome measurements including prehospital return of spontaneous circulation, one-month survival, and one-month neurological outcomes. All OHCA survivors were followed up for up to one month by EMS providers in charge. The etiology of cardiac arrest was determined by the physician in charge who was also responsible for evaluating and reporting neurological outcomes one month after the event based on the Cerebral Performance Category (CPC) scale with five categories: Category 1, good cerebral performance; category 2, moderate cerebral disability; category 3, severe cerebral disability; category 4, coma or vegetative state; and category 5, death/brain death [15]. These data were registered on the FDMA server. EMS providers in charge were required to register the data if they were missing or invalid.

### 2.5. Study Sample

We identified annual meeting dates from 2011 to 2015 of three national academic organizations—the Japanese Association for Acute Medicine (JAAM), the Japanese Circulation Society (JCS), and the Japanese Society of Intensive Care Medicine (JSICM)—because medical staff members of these organizations were considered to play a critical role in caring for cardiac arrest patients admitted to hospitals. Each meeting was held for three consecutive days, usually every March, October, and February, respectively; however, the 2011 JCS meeting was an exception, as it was held in August for two consecutive days because the Great East Japan Earthquake had occurred that year. We defined the “during meeting dates” group as patients admitted after suffering from OHCA during the dates of these academic meetings. Meanwhile, the patients admitted with OHCA during the identical weekdays in the weeks before and after the meetings were defined as the “before meeting dates” and “after meeting dates” groups, respectively [9,10]. For example, the annual JCS meeting of 2015 was held from Friday, 24 April through to Sunday, 26 April; the before-meeting-dates and after-meeting-dates groups were defined as patients admitted from Friday, 17 April through to Sunday, 19 April and Friday, 1 May through to Sunday, 3 May, respectively. In this study, we enrolled all adult patients 18 years old or older for whom resuscitation was attempted by EMS personnel after OHCA, which had occurred before EMS arrival, and who were subsequently transported to hospitals from 1 January 2011 through to 31 December 2015. Patients who received only CPR rescue breathing and those with data missing on bystander CPR or initial documented rhythm were excluded. As the national JCS and JSICM meeting dates were relatively close, we also excluded patients whose group assignment we could not determine.

To study differences in outcomes by meeting location, we divided the cohort by whether the meeting had been held in the Tokyo metropolitan area or other places. Of the 15 national congress meetings held over the five-year period, eight meetings were held in the Tokyo metropolitan area (either Tokyo or Yokohama—two JCS meetings and three JAAM and JSICM meetings, respectively). In addition, the cohort was divided into better or worse outcome areas by median one-month survival with favorable neurological outcomes after OHCA, which was 1.9%, during the period from 2011 to 2015, as there was regional variation in favorable outcomes in patients with OHCA across prefectures in Japan [14].

### 2.6. Outcome Measures

Our primary outcome measure was favorable neurological outcome one month after cardiac arrest, which was defined as CPC scale scores of 1 or 2 [15]. The secondary outcome measure was one-month survival.

### 2.7. Statistical Analysis

Data are presented as numbers (percentages) for categorical variables and median (interquartile range) for continuous variables. The chi-square test was used for binary variables, and the Kruskal–Wallis test was used for continuous and categorical variables to calculate *p*-values. To compare the primary and secondary outcomes among the three groups, we applied a multivariate logistic regression model defining the during-meeting-dates group as the reference group. We obtained adjusted odds ratios (ORs) and their 95% confidence intervals (CIs) after adjusting the following covariables: Age (18–39, 40–64, or ≥65); gender; type of bystander witness (none, family member, or other); type of bystander-initiated CPR (none, compression-only CPR, or conventional CPR); etiology of cardiac arrest (cardiac origin or noncardiac origin); initial documented rhythm (ventricular fibrillation [VF]/pulseless VF or pulseless electrical activity/asystole); public-access AED use; prehospital intravenous fluid; prehospital administration of epinephrine; prehospital advanced airway management; time interval from call to hospital arrival (in one-minute increments); regional variation (better vs. worse outcome areas); and national academic meeting location (Tokyo metropolitan area or others) [11,16]. A subgroup analysis was conducted, stratifying patients according to whether the meeting had been held in the Tokyo metropolitan area or other areas. Next, the patients were divided by whether the OHCA had occurred in better or worse outcome areas to examine regional variation. We conducted sensitivity analysis using alternative definitions of the before- and after-meeting-dates groups: Two, three, and four weeks before and two, three, and four weeks after meeting dates, respectively, instead of one week. We also examined excluded subjects who suffered cardiac arrest during, before, and after the meeting dates of the JCS meeting in 2011 when the Great East Japan Earthquake had occurred. A two-tailed *p*-value of <0.05 was regarded as statistically significant. All analyses were performed with IBM SPSS Statistics 25 (IBM SPSS, Chicago, IL, USA).

## 3. Results

### 3.1. Patient Characteristics

During the study period, 629,471 OHCA patients were registered. Of those, we assessed 554,126 adult patients for whom resuscitation was attempted for eligibility. We excluded 513,277 patients for not being admitted on eligible days, receiving only CPR rescue breathing, unknown information on bystander CPR, or initial documented rhythm, and occurring on 8 or 9 March 2013, which were both before the JCS meeting dates and after the JSICM meeting dates. As a result, we included 40,849 subjects in our analyses, with 14,490 patients in the during-meeting-dates group, 13,518 patients in the before-meeting-dates group, and 12,841 patients in the after-meeting-dates group (Figure 1). Basic characteristics were almost similar except for initial documented rhythm and proportions of patients in the three groups for the national academic meeting locations (Table 1).

### 3.2. Comparison of Outcomes between the Three Groups

For primary outcome components, we found no differences in one-month favorable neurological outcomes among patients admitted during, before, and after meeting dates (Table 2). We found no differences in one-month favorable neurological outcomes among the three groups in multivariate logistic regression analysis (adjusted OR (95% CI) of the before- and after-meeting-dates groups; 1.03 (0.83–1.28) and 1.01 (0.81–1.26), respectively, considering the during-meeting-dates group as the reference; Table 2). Furthermore, no differences were found regarding one-month survival as a secondary outcome (Table 2).

### 3.3. Subgroup Analysis

A subgroup analysis, stratifying patients by better or worse outcome areas and dividing the meeting location into the Tokyo metropolitan area and others, respectively, is shown in Table 3. There were no differences in favorable neurological outcomes among the three groups.

### 3.4. Sensitivity Analysis

When looking at alternative definitions of the before- and after-meeting-dates groups, no differences were observed among the three groups regarding favorable neurological outcomes (Table 4). Similar results were obtained when we excluded subjects who suffered cardiac arrest during, before, and after the JCS meeting dates in 2011 (data not shown).

## 4. Discussion

In this nationwide database study, we investigated the “national meeting effect” in patients with OHCA, comparing data among three groups of patients treated during, before, and after national academic meeting dates, considering regional variation of outcomes and meeting locations. No significant differences in favorable neurological outcomes were observed among the three groups.

Previously, Kitamura et al. reported no differences in adult patients’ post-OHCA outcomes between meeting and nonmeeting dates during the period from 2005 to 2012 in Japan [11]. Another study also suggested that there was little or no “national meeting effect” among patients with AMI in Japan [20]. However, lower 30-day mortality rates were found among high-risk patients with heart failure or cardiac arrest and AMI admitted to major teaching hospitals during relevant national academic meetings in the United States [10,12]. In another study, increased discard rates were found for kidneys procured during American Transplant Congress meetings because of limited staffing [21].

One of the differences between these studies is selection of the control group, which we divided into the before-meeting-dates group and the after-meeting-dates group to investigate the possible positive impact of a national academic meeting on patient outcomes rather than investigating the hypothesis of worse outcomes during the meeting dates. In addition, we examined the “national meeting effect” using a different study period from the previous study and accounted for regional variation of outcomes and meeting places, which might affect the number and composition of physicians who remain to treat patients or who were on call. Contrary to our hypothesis, no significant differences were observed among the three groups.

Generally, annual academic meetings are considered to positively affect patient outcomes by enhancing knowledge of medical staff and performance and promoting research networking and development of guidelines [5]. On the other hand, resuscitation team organization may be different during national meeting dates compared to nonmeeting dates, as many medical personnel attend the relevant meetings. A team approach with strong leadership during resuscitation has been reported to be associated with better outcomes in patients after OHCA as well as in-hospital cardiac arrest [22,23]. Our results suggest that there would be a small impact on patient outcomes as a “post-meeting effect” as such a short time period [3]. In addition, it is speculated that each hospital struggled to provide consistent high-quality care through proper organization of medical staff during meeting dates, regardless of the place where the meeting was held in Japan. These results were the same as those previously described among patients after trauma [24].

Several important limitations in this study should be acknowledged. First, a weak point of this study was the lack of data on in-hospital treatment, including whether extracorporeal CPR, percutaneous coronary intervention, and therapeutic hypothermia management were performed, which are relevant clinical factors that could affect patient outcomes [25,26,27]. Second, prognostic factors after hospital arrival, such as lactate clearance, pupillary light reflex, or pattern of amplitude-integrated electroencephalography, were unavailable [28,29,30]. Third, we could not assess detailed information on the difference in medical staff or organization of resuscitation teams at each hospital among the three groups, considering whether the meeting had been held in the Tokyo metropolitan area or other areas. Fourth, use of long-term outcomes rather than outcomes at one month might alter our results, reflecting much more in-hospital intervention [31]. Fifth, the types of hospitals to which the patients had been admitted were unknown [10,12,32]. Lastly, we could not obtain data on comorbidities, which could be possible confounders. Finally, our results cannot be generalized to other countries with different healthcare system settings, as we only focused on national academic meetings in Japan.

## 5. Conclusions

We found no differences in favorable neurological outcomes among OHCA patients admitted before, during, and after relevant national academic meeting dates, even after considering regional variation of outcomes across Japan and meeting locations. Meeting attendance is important for medical professionals to network and share new knowledge, and for hospitals to strive to deliver equitable, high-quality care during meeting dates.

## Figures and Tables

**Figure 1 ijerph-16-05130-f001:**
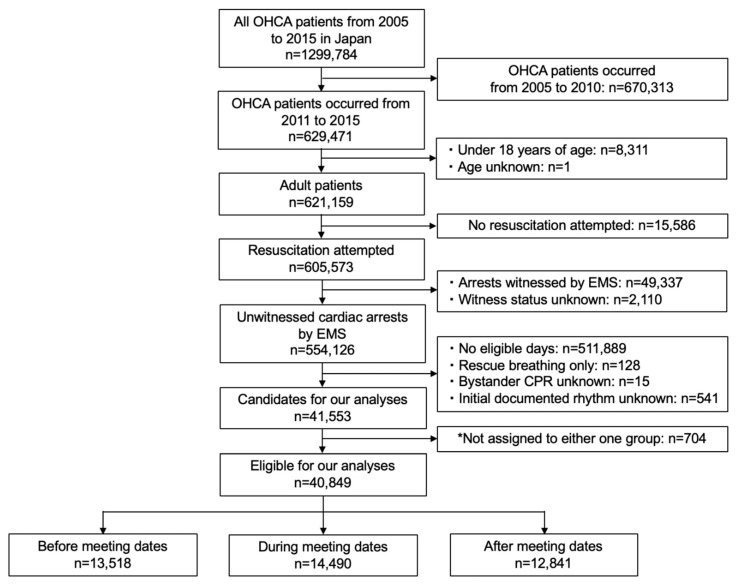
Flow diagram of the study population. * As the national JCS and JSICM meeting dates were relatively close, we excluded patients whose group assignment we could not determine (i.e., during-, before-, or after-the-meeting-dates group). OHCA: Out-of-hospital cardiac arrests; EMS: Emergency medical services; CPR: Cardiopulmonary resuscitation; JCS: Japan Circulation Society; JSICM: Japanese Society of Intensive Care Medicine.

**Table 1 ijerph-16-05130-t001:** Characteristics of OHCA patients admitted during, before, and after national academic meeting dates.

	Before Meeting Dates Group *n* = 13,518	During Meeting Dates Group *n* = 14,490	After Meeting Dates Group *n* = 12,841	*p*-Value
Age (years), median (IQR)	79 (68, 86)	79 (68, 77)	79 (67, 86)	0.131
18–39 (years), *n* (%)	478 (3.5)	533 (3.7)	472 (3.7)	0.564
40–64 (years), *n* (%)	2199 (16.3)	2395 (16.5)	2177 (17.0)	
≥65 (years), *n* (%)	10,841 (80.2)	11,562 (79.8)	10,192 (79.4)	
Male, *n* (%)	7691 (56.9)	8175 (56.4)	7240 (56.4)	0.638
Type of bystander witness				
None, *n* (%)	8807 (65.2)	9475 (65.4)	8288 (64.5)	0.411
Family member, *n* (%)	2798 (20.7)	3044 (21.0)	2736 (21.3)	
Other, *n* (%)	1913 (14.1)	1971 (13.6)	1817 (14.2)	
Type of bystander-initiated CPR				
None, *n* (%)	6983 (51.7)	7487 (51.7)	6566 (51.1)	0.825
Chest compression-only CPR, *n* (%)	5529 (40.9)	5956 (41.1)	5325 (41.5)	
Conventional CPR, *n* (%)	1006 (7.4)	1047 (7.2)	950 (7.4)	
Initial documented rhythm				
VF/pVT, *n* (%)	947 (7.0)	1026 (7.1)	998 (7.8)	0.031
PEA/Asystole, *n* (%)	12,571 (93.0)	13,464 (92.9)	11,843 (92.2)	
Etiology				
Cardiac, *n* (%)	8135 (60.2)	8718 (60.2)	7812 (60.8)	0.444
Other, *n* (%)	5383 (39.8)	5772 (39.8)	5029 (39.2)	
Shocks by public access-AED use, *n* (%)	142 (1.1)	148 (1.0)	151 (1.2)	0.431
Intravenous fluid, *n* (%)	4344 (32.1)	4687 (32.3)	4119 (32.1)	0.880
Epinephrine administration, *n* (%)	2193 (16.2)	2275 (15.7)	2058 (16.0)	0.443
Advanced airway management, *n* (%)	7196 (53.2)	7699 (53.1)	6967 (54.3)	0.128
Time interval from call to hospital arrival (in one-minute increments), median (IQR)	31 (26, 39)	32 (26, 39)	31 (26, 39)	0.131
Better outcome areas, *n* (%)	6509 (48.2)	6990 (48.2)	6255 (48.7)	0.620
Worse outcome areas, *n* (%)	7009 (51.8)	7500 (51.8)	6.586 (51.3)	
National academic meeting place				
Tokyo metropolitan area, *n* (%)	7236 (53.5)	7142 (49.3)	6980 (54.4)	<0.001
Other area, *n* (%)	6282 (46.5)	7348 (50.7)	5861 (45.6)	

OHCA, out-of-hospital cardiac arrest; IQR, interquartile range; CPR, cardiopulmonary resuscitation; VF, ventricular fibrillation; pVT, pulseless ventricular tachycardia; PEA, pulseless electrical activity; AED, automated external defibrillator. *p*-values were calculated using chi-square test for binary variable and Kruskal–Wallis test for continuous and categorical variables.

**Table 2 ijerph-16-05130-t002:** Outcomes of OHCA patients during, before, and after the meeting dates.

	Before-Meeting-Dates Group	During-Meeting-Dates Group	After-Meeting-Dates Group
Favorable neurological outcome, % (*n*/*N*)	1.6 (211/13,518)	1.7 (243/14,490)	1.8 (232/12,841)
Crude OR (95% CI)	0.93 (0.77–1.11)	Reference	0.87 (0.72–1.05)
Adjusted OR (95% CI)	0.95 (0.78–1.15)	Reference	1.06 (0.88–1.28)
One-month survival, % (*n/N*)	3.8 (516/13,518)	3.8 (555/14,490)	3.8 (493/12,841)
Crude OR (95% CI)	1.00 (0.88–1.13)	Reference	0.99 (0.88–1.13)
Adjusted OR (95% CI)	1.00 (0.88–1.14)	Reference	0.98 (0.86–1.11)

OHCA, out-of-hospital cardiac arrest; OR, odds ratio; CI, confidence interval. Adjusted OR and their 95% CIs were obtained after adjusting for age; gender; type of bystander witness; type of bystander-initiated CPR; origin of arrest; initial documented rhythm; public-access AED use; prehospital intravenous fluid; prehospital administration of epinephrine; prehospital advanced airway management; time interval from call to hospital arrival; regional variation; and national academic meeting location.

**Table 3 ijerph-16-05130-t003:** Favorable neurological outcomes among the three groups with stratification for better or worse outcome areas and dividing the meeting location into the Tokyo metropolitan area or other areas, respectively.

**Overall**				
	**Meeting Location**	**Before-Meeting-Dates Group**	**During-Meeting-Dates Group**	**After-Meeting-Dates Group**
Favorable neurological outcome, % (*n/N*)	Overall	1.6 (211/13,518)	1.7 (243/14,490)	1.8 (232/12,841)
	Tokyo metropolitan area	1.6 (112/7236)	1.6 (115/7142)	1.7 (121/6980)
	Others	1.6 (99/6282)	1.7 (128/7348)	1.9 (111/5861)
Adjusted OR for favorable neurological outcome	Overall	0.95 (0.78–1.15)	Reference	1.06 (0.88–1.28)
	Tokyo metropolitan area	1.05 (0.74–1.48)	Reference	0.88 (0.62–1.25)
	Others	0.97 (0.68–1.39)	Reference	1.13 (0.80–1.60)
**Better Outcome Areas**				
	**Meeting Location**	**Before-Meeting-Dates Group**	**During-Meeting-Dates Group**	**After-Meeting-Dates Group**
Favorable neurological outcome, % (*n/N*)	Overall	2.0 (132/6509)	2.1 (145/6990)	2.1 (133/6255)
	Tokyo metropolitan area	2.1 (74/3491)	2.0 (71/3512)	2.0 (67/3392)
	Others	1.9 (58/3018)	2.1 (74/3478)	2.3 (66/2863)
Adjusted OR for favorable neurological outcome	Overall	1.00 (0.78–1.28)	Reference	1.01 (0.83–1.25)
	Tokyo metropolitan area	1.05 (0.74–1.48)	Reference	0.88 (0.62–1.25)
	Others	0.97 (0.68–1.39)	Reference	1.13 (0.80–1.60)
**Worse Outcome Areas**				
	**Meeting Location**	**Before-Meeting-Dates Group**	**During-Meeting-Dates Group**	**After-Meeting-Dates Group**
Favorable neurological outcome, % (*n/N*)	Overall	1.1 (79/7009)	1.3 (98/7500)	1.5 (99/6586)
	Tokyo metropolitan area	1.0 (38/3745)	1.2 (44/3630)	1.5 (54/3588)
	Others	1.3 (41/3264)	1.4 (54/3870)	1.5 (45/2998)
Adjusted OR for favorable neurological outcome	Overall	1.15 (0.86–1.55)	Reference	0.93 (0.73–1.20)
	Tokyo metropolitan area	0.82 (0.52–1.29)	Reference	1.25 (0.82–1.89)
	Others	0.90 (0.59–1.37)	Reference	1.07 (0.71–1.62)

OR, odds ratio; CI, confidence interval; Adjusted OR and their 95% CIs were obtained after adjusting for age; gender; type of bystander witness; type of bystander-initiated CPR; origin of arrest; initial documented rhythm; public-access AED use; prehospital intravenous fluid; prehospital administration of epinephrine; prehospital advanced airway management; and time interval from call to hospital arrival.

**Table 4 ijerph-16-05130-t004:** Favorable neurological outcomes among the three groups with alternative definitions of the before- and after-meeting-dates groups.

	Before Meeting Dates Group	During Meeting Dates Group	After Meeting Dates Group
±2 ^a^			
Favorable neurological outcome, % (*n/N*)	1.7 (223/12,948)	1.7 (216/12,537)	1.8 (226/12,366)
Adjusted OR (95% CI)	0.91 (0.71–1.17)	Reference	0.96 (0.75–1.23)
±3 ^b^			
Favorable neurological outcome, % (*n/N*)	1.5 (196/13,497)	1.8 (219/12,429)	1.8 (235/12,820)
Adjusted OR (95% CI)	0.76 (0.60–0.96)	Reference	0.97 (0.78–1.22)
±4 ^c^			
Favorable neurological outcome, % (*n/N*)	1.5 (193/13,106)	1.7 (234/13,792)	1.7 (203/12,111)
Adjusted OR (95% CI)	1.01 (0.78–1.30)	Reference	0.99 (0.88–1.12)

^a^ Two weeks before and after meeting dates as the before-meeting-dates group and after-meeting-dates group. ^b^ Three weeks before and after meeting dates as the before-meeting-dates group and after-meeting-dates group. ^c^ Four weeks before and after meeting dates as the before-meeting-dates group and after-meeting-dates group. OR, odds ratio; CI, confidence interval. Adjusted OR and their 95% CIs were obtained after adjusting for age; gender; type of bystander witness; type of bystander initiated CPR; origin of arrest; initial documented rhythm; public-access AED use; prehospital intravenous fluid; prehospital administration of epinephrine; prehospital advanced airway management; time interval from call to hospital arrival; regional variation; and national academic meeting location.

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
