# Peer review of "Geographical Differences and the National Meeting Effect in Patients with Out-of-Hospital Cardiac Arrests: A JCS–ReSS Study Report"

_ijerph, 2019, doi:10.3390/ijerph16245130_

Round 1

Reviewer 1 Report

The authors analyzed the international database. The methods are compatible with previous studies and well organized and conducted. 

However, some critical points should be pointed out. 

 Attendances of medical professions in Medical meetings can affect not prehospital factors but the in-hospital factors. However, the authors didn't cover any hospital factors such as PCI or E-CPR.  I can not understand why authors only divided regions of Japan into East and West. Prehospital factors look much different between prefectures even within the Kanto region (Okuboa et al. 2018). I think that the characteristics of Hokkaido and Northeast of the main island of Japan are much different from those of Tokyo cannot be tied together as eastern Japan. I can not understand the main hypothesis, 'patient outcomes would be better after the meeting dates than during the meeting dates because of reduced staffing during the meeting dates and the academic meeting’s positive impact on physician performance'. Previous studies had been failed to show the harmful effects of reduced staffing during medical meeting dates. Academic meeting's emphatic impact is vague and not scientific. Authors didn't provide any evidence and previous quality improvement studies showed, a lot of time and effort are needed for improving clinical outcomes after announcements of guidelines or education courses. 

I cannot find meaningful additional information from this study from the 2016 article by Kitamura et al. For overcoming limitations, in-hospital variables should be covered. Also specified and clinically meaningful regional analysis should be conducted. Urban vs. rural, better vs. worse outcome areas, sufficient vs. insufficient medical resources prefectures and distances from the meeting area can be examples. 

Author Response

Response to Reviewer 1 comments

Point 1: Attendances of medical professions in Medical meetings can affect not prehospital factors but the in-hospital factors. However, the authors didn't cover any hospital factors such as PCI or E-CPR. 

Response 1: We completely agree with your concern. We should have adjusted in logistic regression analysis by adding in-hospital parameters or interventions such as GCS, lactate, pupil diameter/reflex on arrival and PCI or ECPR. However, as the database we used was based on the Utstein-style reporting format, these data were unavailable, which is definitely the major limitation in this study. As so, we have emphasized this weak point in the limitations.

Revision: Several important limitations in this study should be acknowledged. First, a weak point of this study was the lack of data on in-hospital treatment including whether extracorporeal CPR, percutaneous coronary intervention, and therapeutic hypothermia management was performed, which are relevant clinical factors that could affect patient outcomes [25-27]. Second, prognostic factors after hospital arrival such as lactate clearance, pupillary light reflex, or pattern of amplitude-integrated electroencephalography were unavailable [28-30].

Point 2: I can not understand why authors only divided regions of Japan into East and West. Prehospital factors look much different between prefectures even within the Kanto region (Okuboa et al. 2018). I think that the characteristics of Hokkaido and Northeast of the main island of Japan are much different from those of Tokyo cannot be tied together as eastern Japan. 

Response 2: Actually, there is no rationale for dividing Japan just into East and West. Instead we divided Japan into better vs. worse outcome areas based on the work by Okubo et al. in accordance with your suggestions.

Revision: Also, the cohort was divided into better or worse outcome areas by median one-month survival with favorable neurological outcomes after OHCA in 2014, as there was regional variation in favorable outcomes in patients with OHCA across prefectures in Japan [Okubo, et al, 2018].

Point 3: I can not understand the main hypothesis, 'patient outcomes would be better after the meeting dates than during the meeting dates because of reduced staffing during the meeting dates and the academic meeting’s positive impact on physician performance'. Previous studies had been failed to show the harmful effects of reduced staffing during medical meeting dates. Academic meeting's emphatic impact is vague and not scientific. Authors didn't provide any evidence and previous quality improvement studies showed, a lot of time and effort are needed for improving clinical outcomes after announcements of guidelines or education courses. 

Response 3: As you pointed out, this hypothesis is just our speculation without any evidence and beyond theory. We have deleted this speculation and revised accordingly.

Point 4: I cannot find meaningful additional information from this study from the 2016 article by Kitamura et al. For overcoming limitations, in-hospital variables should be covered. Also specified and clinically meaningful regional analysis should be conducted. Urban vs. rural, better vs. worse outcome areas, sufficient vs. insufficient medical resources prefectures and distances from the meeting area can be examples. 

Response 4: We have addressed this matter by revising limitations and conducting alternative regional analysis as mentioned above.

Reviewer 2 Report

Yumoto and colleagues report herein the results of a Japanese nationwide study of the impact of national scientific meeting on the outcomes of sudden cardiac arrest. They show that the “national meeting effect” is non-existent for survivors of out-of-hospital cardiac arrest. This data is very timely and would be of a major interest to readers. However, there are some minor issues that the authors should revise in the manuscript.

Major Comments to the Author

Methods

In the Statistical Analysis, the authors fail to adjust for clinical predictors of sudden cardiac arrest. Can they elaborate on why these have not been included in the models? Moreover, the use of logistic regression herein raises some concerns. Given that the secondary outcome (30-day survival) is a time-varying variable, could the authors comment on why they have not used a Cox regression method?

Discussion

Given the short follow-up time (1 month), could it be that impact of the meeting might be more apparent after 1 month of follow-up? It would be very interesting if the authors could incorporate this in their discussion and comment on how it affects the interpretation of their results. I think this could also be acknowledged in the limitations.

Minor Comments to the Author

Contractions should be avoided in the manuscript – the authors should correct the “study’s” on lines 28 and 290 The Abstract is supposed to provide a summary of the study, not a detailed description of the procedures and protocols used. Could the authors revise this section, ensuring that the methodological details are very succinct. For the Flow diagram, the 2nd “Resuscitation attempted” box should be revised to reflect the fact that these were unwitnessed cardiac arrests. In Table 2, the order of the outcomes should be changed, with the primary outcome presented first, before the secondary outcome.

Author Response

Response to Reviewer 2 comments

Major Comments to the Author

Methods

Point 1: In the Statistical Analysis, the authors fail to adjust for clinical predictors of sudden cardiac arrest. Can they elaborate on why these have not been included in the models? Moreover, the use of logistic regression herein raises some concerns. Given that the secondary outcome (30-day survival) is a time-varying variable, could the authors comment on why they have not used a Cox regression method?

Response 1: Ideally, we should have considered other clinical parameters in logistic regression model including Glasgow Coma Scale, pupil reflex/index, lactate level or findings of aEEG…, plus in-hospital intervention such as with or without PCI and ECPR. However, these data were unavailable as this database was Utstein-style reporting of OHCA. Similarly, outcomes were available only at the time of 1 month after the event as a categorical variable. The manuscript has been revised to emphasize these weak points in the limitations.

Revision: Several important limitations in this study should be acknowledged. First, a weak point of this study was the lack of data on in-hospital treatment including whether extracorporeal CPR, percutaneous coronary intervention, and therapeutic hypothermia management was performed, which are relevant clinical factors that could affect patient outcomes [25-27]. Second, prognostic factors after hospital arrival such as lactate clearance, pupillary light reflex, or pattern of amplitude-integrated electroencephalography were unavailable [28-30].

Discussion

Point 2: Given the short follow-up time (1 month), could it be that impact of the meeting might be more apparent after 1 month of follow-up? It would be very interesting if the authors could incorporate this in their discussion and comment on how it affects the interpretation of their results. I think this could also be acknowledged in the limitations.

Response 2: Long-term outcome should also be looked at and would be definitely interesting, however, this data are unavailable from the nature of database study. We have discussed together in the limitations.

Revision: Fourth, use of long-term outcomes rather than outcomes at one-month might alter our results reflecting much more in-hospital intervention [31].

Minor Comments to the Author

Point 3: Contractions should be avoided in the manuscript – the authors should correct the “study’s” on lines 28 and 290.

Response 3: These have been corrected accordingly.

Point 4: The Abstract is supposed to provide a summary of the study, not a detailed description of the procedures and protocols used. Could the authors revise this section, ensuring that the methodological details are very succinct.

Response 4: We have revised abstract as more brief description.

Point 5: For the Flow diagram, the 2nd “Resuscitation attempted” box should be revised to reflect the fact that these were unwitnessed cardiac arrests.

Response 5: Figure 1 has been revised appropriately.

Point 6: In Table 2, the order of the outcomes should be changed, with the primary outcome presented first, before the secondary outcome. 

Response 6: The order has been changed accordingly.

Round 2

Reviewer 1 Report

The regional analysis of your study looks incomplete.
 - Detailed explanation for better and worse outcome areas should be added in the Materials and Methods. By mean or median? By rank?
 - It looks that better and worse groups are classified according to the 2014 outcome only. Your research includes data from 2005 to 2015.

- Why you didn't analysis for each prefecture?

In Table 3, "west japan" remains undeleted.

Author Response

Response to Reviewer 1 comments

The regional analysis of your study looks incomplete.

 Point 1: Detailed explanation for better and worse outcome areas should be added in the Materials and Methods. By mean or median? By rank?

Point 2: It looks that better and worse groups are classified according to the 2014 outcome only. Your research includes data from 2005 to 2015.

Response: We address these concerns together. Better and worse outcome areas have been split by median 1-month favorable outcome after cardiac arrest, which was 1.9%, during the period from 2011 to 2015. An original file contains data from 2005 to 2015, though this study included OHCA patients which had occurred from 2011 to 2015. That’s why we split the data as above based on the data from 2011 to 2015. All the data have been reanalyzed and tables have been revised as appropriately.

Revision: Also, the cohort was divided into better or worse outcome areas by median one-month survival with favorable neurological outcomes after OHCA, which was 1.9%, during the period from 2011 to 2015, as there was regional variation in favorable outcomes in patients with OHCA across prefectures in Japan [14].

Point 3: Why you didn't analysis for each prefecture?

Response: That analysis should be interesting. But a number of 1-month favorable outcomes in during, before, and after meeting groups were 243, 211, and 232, respectively. When we look at 47 individual prefectures, it would be impossible to analyze statistically due to its small samples each. We thought it reasonable to just split better and worse outcome areas as you have kindly suggested.

Point 4: In Table 3, "west japan" remains undeleted.

Response: “West Japan” has been corrected to “worse outcome areas”.

Round 3

Reviewer 1 Report

Concerns are well covered.